# Cut-to-Length Harvesting Options for the Integrated Harvesting of the European Industrial Poplar Plantations

**Raffaele Spinelli** [1,2,3], **Natascia Magagnotti** [1,2,3,*], **Fabio De Francesco** [1], **Barnabáš Kováč** [4], **Patrik Heger** [4], **Dávid Heilig** [5,6], **Bálint Heil** [5,6], **Gábor Kovács** [5,6] **and Tomáš Zemánek** [7]

1   Consiglio Nazionale delle Ricerche-Istituto per la Bioeconomia (CNR IBE), Via Madonna del Piano 10, 50019 Sesto Fiorentino, Italy
2   Forest Research Institute, University of the Sunshine Coast, Locked Bag 4, Maroochydore, Queensland 4558, Australia
3   Forestry Research Institute of Sweden (SKOGFORSK), Dag Hammarskjölds Väg 36A, 751 83 Uppsala, Sweden
4   IKEA Industry Slovakia, Továrenská 2914/19, 901 01 Malacky, Slovakia
5   Institute of Environmental and Nature Protection, University of West Hungary, POB 132, H-9401 Sopron, Hungary
6   Ökoforestino Ltd., Ibolya út 11. V/21., H-9400 Sopron, Hungary
7   Department of Engineering, Faculty of Forestry and Wood Technology, Mendel University in Brno, 613 00 Brno, Czech Republic
*   Correspondence: natascia.magagnotti@ibe.cnr.it

**Abstract:** Plantation forestry has a long history in Europe and still supports local industry, generating employment, improving environmental quality, and mitigating climate change. As part of these plantations, medium-rotation poplars (5–8 years) provide good quality logs for fiber production, and the branches and tops can be converted into green energy. Finding a cost-effective harvesting system for this plantation is challenging due to the small tree size and the need for log production, which prevents whole-tree chipping. To verify the economic benefit of using small mechanized cut-to-length (CTL) technology, four different CTL chains were tested in western Slovakia. All chains tested consisted of a harvester and a forwarder. Each machine had a skilled operator and was timed while cutting and processing (or forwarding) eight experimental sample plots. Sample plots were randomly assigned to each treatment, and each one covered an area between 0.08 and 0.10 ha (120–170 trees). Harvester productivity ranged from 2.2 to 4.2 bone-dry tons per scheduled machine hour (BDT SMH$^{-1}$), and harvester cost from EUR 11 to EUR 22 BDT$^{-1}$. Forwarding productivity and cost ranged from EUR 2.0 to 4.5 BDT SMH$^{-1}$ and from EUR 9 to 20 BDT$^{-1}$. Total harvesting costs ranged between EUR 26 and 36 BDT$^{-1}$. Choosing a smaller harvester is preferable due to the small tree size, which caps productivity regardless of a machine's intrinsic potential. Furthermore, small harvesters and forwarders are lighter on the ground, which can be a valuable asset on the many wet sites planted with poplar.

**Keywords:** harvester; forwarder; time study; productivity; cost

## 1. Introduction

Plantation forests have been expanding since the late 1960s, driven by rapid population growth and the increasing global demand for timber products, [1] and may currently account for 70% of the world's production of industrial roundwood [2]. Their extraordinary success is due to many factors, especially the remarkable efficiency of concentrating production in a relatively small area, given that they currently represent less than 2% of global land use [3]. In addition to using relatively little land, these crops do not compete with conventional food crops because they are generally established on marginal sites and grow a valuable product while improving the environment at the same time [4,5]. The efficiency of plantation forestry is further increased by industrialization [6], optimized

professional management [7], and the selection of sites characterized by favorable climatic conditions and low land and labor cost, often found in the subtropical regions of the Southern Hemisphere [8].

Europe also has a long experience with plantation forestry, but until recently, the characteristics of European plantations were much different from those of the modern tree farms established in the Southern Hemisphere [9–12]. Most post-war projects focused on covering degraded mountain sites with softwood trees [13], while later efforts directed at surplus farmland were geared to the production of low-grade industrial feedstock within very short rotations and emulated conventional farm crops [14]. While quite successful in their own time, both afforestation models seem to have run their course [15]. Currently, interest has shifted to medium-rotation tree farms, established on agricultural land and capable of producing a mix of timber and biomass, much like those plantations that are so popular in the Southern Hemisphere. Of course, the temperate regions of Europe offer different climate and soil fertility conditions compared with the subtropical regions of Africa, Asia, and South America, and therefore European plantations are established with different species and offer lower yields. Nevertheless, they may provide significant benefits in terms of additional revenue for the farmers, reduction of site impacts, diversification of the agricultural landscape, and better control of the raw material supply for the local wood industry [16,17].

In Europe, medium-rotation industrial plantations offer a new opportunity for poplar nurseries and growers. Placed between traditional poplar stands and short-rotation energy crops, these new plantations are designed to produce a mix of quality logs for fiber production and low-grade tops for energy use. There are currently tens of thousands of hectares of these plantations in Europe—especially in the Eastern regions [18].

Aggressive global competition requires mechanization of most work steps, especially harvesting, which represents over half of the total management cost of forest plantations [7]. Mechanized cut-to-length (CTL) technology is the classic European solution to forest harvesting [19], and it is quite popular in Eastern Europe, too [20–22]. Using CTL technology to harvest medium-rotation poplar offers many benefits, namely, the machines are already available on site, use on medium-rotation poplar may allow faster depreciation of expensive machines, value recovery is equal to or better than that achieved with motor-manual systems, work safety is greatly enhanced, and the CTL harvesting system can manage tighter landings and smaller lots than is normally achieved by the whole-tree harvesting system [23]. On the other hand, CTL technology is designed for handling one tree at a time and encounters a real challenge with the small trees offered by the dense grid, mid-rotation plantations [24].

While new multi-tree harvester heads may help overcome the small tree challenge [25,26], they still represent a small minority within the CTL harvester fleets of most European countries, and the question arises about what conventional CTL machines are best for tackling medium-rotation poplar trees. "Small equipment for small trees" is a simple equation that needs some defining. First of all, equipment downsizing may respond to accessibility and investment limitations more than to tree size constraints. Secondly, one should know how small it is worth going before the selected equipment turns out to be too small.

The goal of this study was to compare four different small CTL equipment chain options for use in a typical medium-rotation poplar plantation and find out which one incurred the lowest harvesting cost and under what specific conditions. The null hypothesis was that of no harvesting cost difference between the four treatments.

## 2. Materials and Methods

### 2.1. Materials: Machines and Operators

The systems under investigation all consisted of the classic CTL combination of a harvester and a forwarder. The harvester felled and processed poplar trees into 4.0 m logs, with a length tolerance ± 0.15 m and a minimum small-end diameter of 8 cm over

bark. The forwarder separately hauled the logs and the biomass (tops and branches) to the roadside landing area.

The harvesters used for the test were Agama AH6, Rottne H8, Sampo HR46, and Vimek 404. All machines were four-wheelers originally designed for thinning operations (Figure 1). Their power ranged from 50 to 125 kW, and their weight from 4.5 to 12 tons. The cutting diameter varied between 30 and 50 cm, all more than adequate for medium-rotation poplar (Table 1).

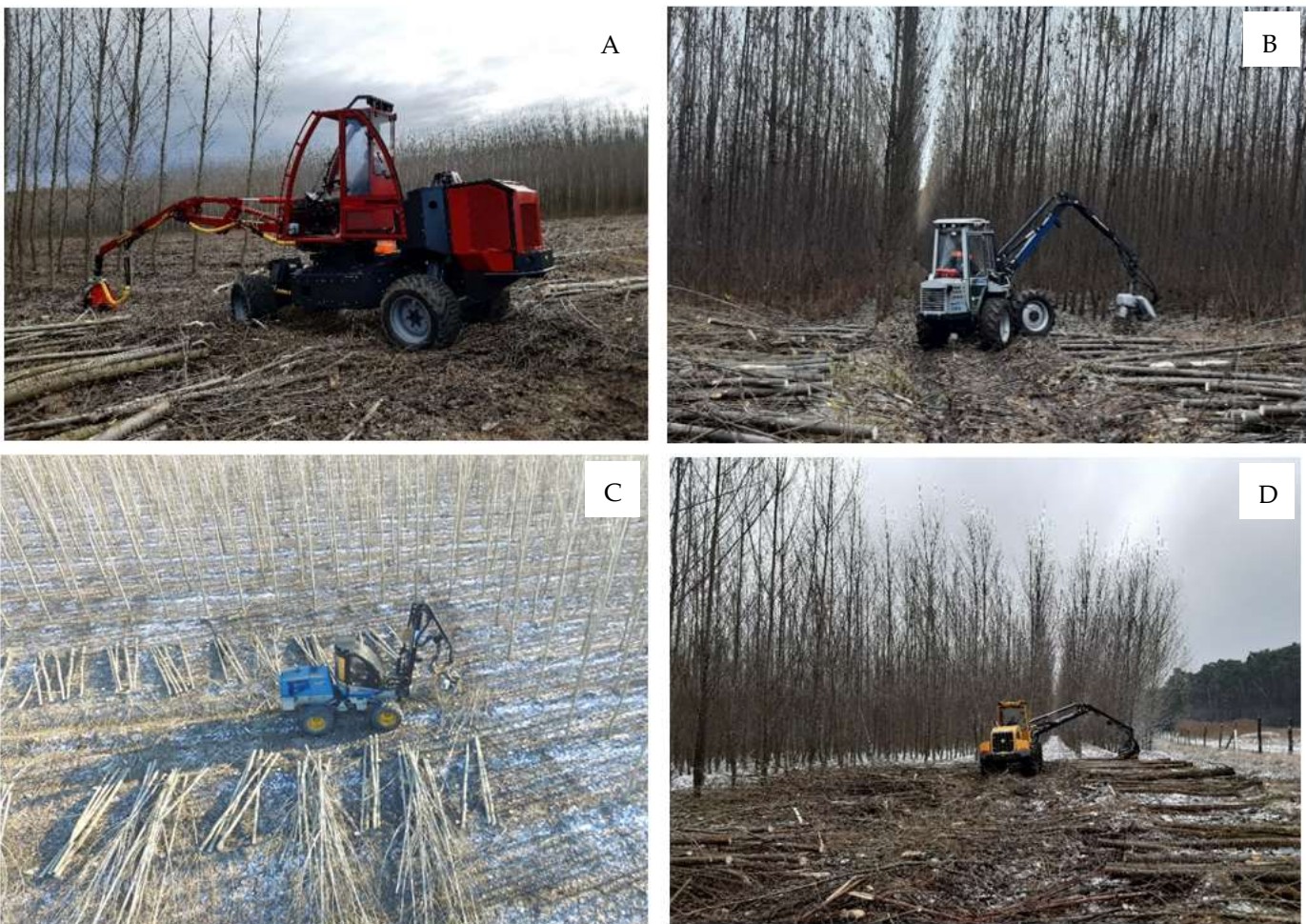

**Figure 1.** The four cut-to-length harvesters on test: Agama AH6 (**A**); Vimek 404 (**B**); Rottne H8 (**C**); Sampo HR46 (**D**).

**Table 1.** Technical specifications of the four harvesters tested.

| Concept | Machine | Power | Weight | Head | Ø Capacity |
|---------|---------|-------|--------|------|-----------|
| Type | Model | kW | kg | type | cm |
| Micro | Vimek 404 | 50 | 4500 | Keto Forst V4 | 34 |
| Hybrid | Agama H6 | 75 | 7700 | Nisula 325H | 34 |
| Small | Sampo HR46 | 124 | 9500 | Kesla 18RH | 50 |
| Small | Rottne H8 | 125 | 10,200 | EGS 406 | 43 |

For the purpose of a logical classification, the harvesters were categorized as follows:

- Micro-harvester (Vimek): the lightest, smallest, and cheapest on the market, with a weight of fewer than 5 tons and a power of only 50 kW.

- Hybrid harvester (Agama): a machine quite similar to the previous one but fitted with an auxiliary electric engine to boost power at the moment of peak demand. Of course, the additional power unit resulted in a slight increase in both weight and price.
- Thinning harvester (Rottne and Sampo): a harvester capable of handling small and medium size trees. Heavier, more powerful, and more expensive than the others, but also more versatile and potentially more productive. Since this type of machine is relatively common, two different models were tested in order to get some idea of the possible variation found within this family of machines.

For consistency, the micro-harvester was associated with a six-wheeled micro-forwarder (Vimek 610-50 kW, 5 t payload capacity), while all the other machines were associated with the same eight-wheeled thinning forwarder (Sampo FR28-124 kW, 10 t payload capacity) (Figure 2).

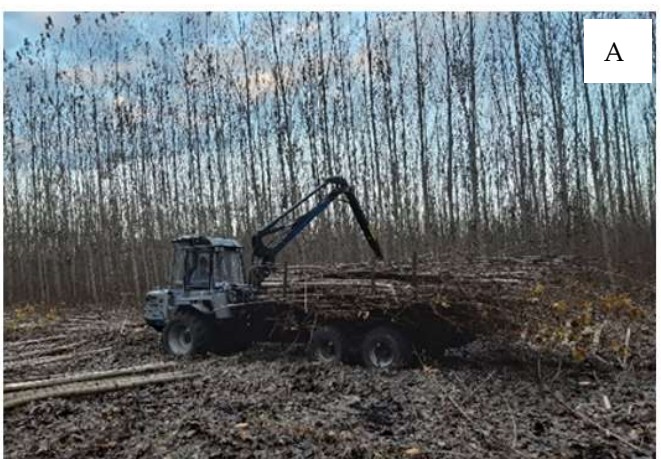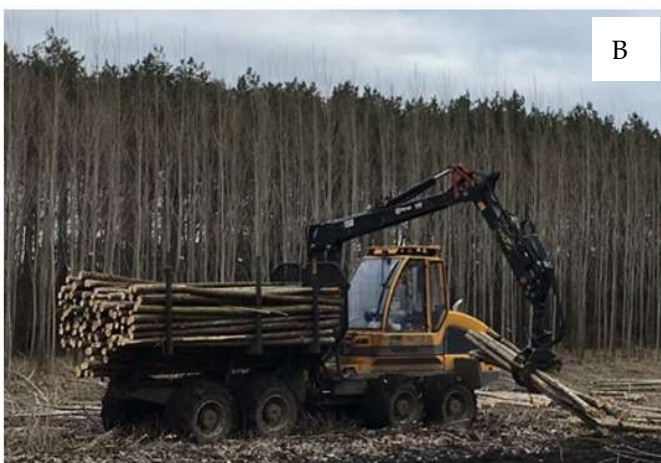

**Figure 2.** The two forwarders on the test: Vimek 610 (**A**) for the Vimek chain; Sampo FR28 (**B**) for all other chains.

Each machine had its own dedicated operator, who was a qualified forestry professional with significant experience in his machines and specific tasks. The operators were selected after observing over a dozen contractors and casting out those who were not considered representative of the regional pool of machine operators. All test operators were informed about the purpose of the test and the methods adopted, and all made their best efforts to cooperate towards the best success of the trial. They were all given at least one full day of practice outside the marked experimental sample plots in order to familiarize themselves with their specific tasks and settings before engaging in the experiment. All harvester head measuring systems were checked and calibrated before starting the trial.

### 2.2. Materials: Test Site—Location and Characteristics of the Plantation

The four alternative chains were tested in the same compartment of the Nivky Plantation, near Veľké Leváre in western Slovakia (48°31′25.11″ N; 17°03′25.21″ E in WGS84) during winter 2020–2021. In particular, the Vimek chain was tested from 9 to 12 November 2020 and all other chains from 8 to 22 February 2021. The test was conducted on a 5-year-old poplar plantation. The plantation was established at a square spacing of 3.0 m × 2.0 m with the hybrid poplar (*Populus x euramericana* Dode (Guinier)) clone 'AF16'. The plantation covered about 10 ha, but the trials were conducted on 3 ha only, while the rest was reserved for practice, fine-tuning, and warm-up before starting the trials.

The mean stocking at the time of cut was 52 bone-dry tons (BDT) per ha. The mean total tree mass (stem and the branches) was 30 dry kg, net of any harvesting losses. The mean diameter at breast height (DBH) was 12 cm, and the mean total height was 15 m. In total, over 5200 poplar trees were harvested during the study, amounting to 155 BDT

of total mass, 60% of which was represented by logs, the rest by energy biomass (tops and branches).

### 2.3. Methods: Experimental Design

Each treatment was assigned 8 sample plots, and each sample plot covered an area between 0.08 and 0.10 ha and contained between 120 and 170 trees. The idea was that each sample plot should require at least 1 work hour to fell and process and as much to extract to the roadside. Furthermore, the amount of wood on each sample plot had to be enough for at least two full forwarder loads—one for the logs and the other for the biomass. The experimental sample plots were randomly assigned to each treatment. Those assigned to the larger harvesters (Rottne and Sampo) were 5 rows wide, while those assigned to the smaller harvesters (Agama and Vimek) were resized to a width of 4 rows to achieve an ideal match between work frontage and crane outreach. All sample plots were about 30 trees deep, which explains why some sample plots had a surface of 0.08 ha (the 4-row sample plots) and others 0.10 ha (the 5-row sample plots). The beginning and the end of each sample plot were clearly marked with high-visibility paint to prevent misunderstanding.

### 2.4. Methods: Determining Mass Output

The DBH of all trees in all sample plots was measured manually with a measuring tape. Furthermore, 6 trees—covering the whole DBH distribution—were destructively sampled in order to obtain their total height, the weight of the logs, and the biomass potentially obtained from them [27,28]. That allowed for building a DBH-height curve and an allometric equation for estimating the mean height and the standing mass on each individual plot [29]. The homogeneity of even-aged clonal poplar makes it possible to build reliable allometric functions with such a small sample [30–32]. Mass estimates were later adjusted using ad-hoc correction factors obtained by matching the total log and biomass estimates with the actual amounts taken to the weighbridge available at the receiving factory. Moisture content (i.e., water mass fraction) was determined both at the time of destructive sampling and at delivery to the factory so as to match dry mass with dry mass. The moisture content at delivery was 57%. The ratio between factory mass and inventory mass was 0.99 for logs and 0.72 for chips (tops and branches), and these ratios were applied to the sample plot inventory in order to correct the estimated mass into the actual mass at the factory. The overall correction factor was 0.86, meaning that the final field inventory overestimated the actual harvest by about 15% (partly due to field losses).

### 2.5. Methods: Determining Time Input

At the time of harvesting, researchers determined for each sample plot: productive time consumption, using a stopwatch accurate to the second; delay time (as above); number of trips (for forwarders only); payload fill rate (actual load as a % of maximum available payload volume, visually estimated—forwarders only); extraction distance (forwarder only—estimated from Google Earth pictures, from the center of the sample plot to the center of the landing). As a safe measure, sample plot-level time consumption was determined concurrently by the researchers on site using stopwatches and by the operators on the machines using their own wristwatches (obviously at a lower definition—one minute). Furthermore, action cameras were attached to all machines in order to acquire a permanent record of operation in case of any doubts or mess-ups. All data used in the final sample plot study derived from the researcher's time study were extremely accurate in both the timing itself and in the separation of work time from delays [33]. As an additional safeguard, researcher records were compared with operator records, and they all presented a reasonably good match, which excluded any gross errors in time data collection.

Occasionally, the mass of wood on one sample plot would exceed the capacity of the forwarder assigned to that sample plot. When that occurred, partial loads were hauled in order not to mix the time and the materials belonging to each sample plot. Therefore, the vehicle travel time was multiplied by the payload fill rate of that load in order to account

for the fact that a full load would have been extracted in real operational conditions, and therefore that same travel time would have been distributed over a larger payload.

Since the study was not long enough to accurately estimate delay time, all estimates were conducted based on the net work time per sample plot, which was inflated by 20% in order to account for preparation and delays. This 20% increase was consistent with the findings of previously published studies, with special reference to the harvesting of plantation forestry [34].

### 2.6. Methods: Determining Machine Cost

Machine cost was assumed to be the rates actually charged by the service providers. These were EUR 40 per scheduled machine hour (SMH) for the Vimek harvester and the two forwarders (Sampo and Vimek), EUR 45 $SMH^{-1}$ for the Agama harvester, EUR 60 $SMH^{-1}$ for the Sampo harvester, and EUR 65 $SMH^{-1}$ for the Rottne harvester. The Authors acknowledge that individual rates can hardly offer a general benchmark and encourage readers operating under very different economic environments to recalculate harvesting costs using their own rates and the productivity data presented in this paper. Ideally, one may use official rates when available. Unfortunately, that was not the case in this study. However, the rates charged by the service providers selected for the study were in line with those charged by other service providers in the region—so even if not official, the selected rates might generally be considered representative of the region.

### 2.7. Methods: Data Analysis

The sample plot study data was used to quantify machine productivity and treatment cost as average values, and the differences between alternative treatments were checked using a general linear model (GLM), which is both accurate and robust against violations of the main statistical assumptions [35]. That procedure was followed by post-hoc tests conducted according to Tukey–Kramer's technique. Furthermore, regression analysis was used to test the effect of field stocking on harvesting cost and log yield separately for each machine system option. For all analyses, the significance level was set at $p < 0.05$.

### 3. Results

Mean field stocking ranged from 43 to 63 BDT $Ha^{-1}$, and mean diameter at breast height from 11 to 13 cm. Statistical analysis showed that field conditions were not statistically different between the four treatments (Table 2).

**Table 2.** Characteristics of the experimental sample plots (mean values).

|  |  | **Agama** | **Rottne** | **Sampo** | **Vimek** | *p* |
|---|---|---|---|---|---|---|
| Plot surface | ha | 0.08 [a] | 0.10 [b] | 0.10 [b] | 0.08 [a] | <0.0001 |
| DBH | cm | 11.9 [a] | 12.2 [a] | 12.2 [a] | 12.4 [a] | 0.1151 |
| Plot mass | BDT | 3.96 [a] | 5.26 [b] | 5.28 [b] | 4.21 [a] | <0.0001 |
| Stocking | BDT ha$^{-1}$ | 49.4 [a] | 52.4 [a] | 52.6 [a] | 53.0 [a] | 0.3761 |
| Log yield | % | 58 [a] | 61 [a] | 61 [a] | 52 [b] | 0.0009 |
| Extraction distance | m | 337 [a] | 339 [a] | 308 [a] | 383 [b] | 0.0291 |

Notes: BDT = Bone-Dry tons (0% water mass fraction); different superscript letters on mean values on the same row indicate a statistically significant difference at the 5% level.

In contrast, the sample plot size was significantly smaller for the two smallest machines since these machines would work on a 4-row frontage instead of a 5-row frontage, and therefore the same sample plot length corresponded to a smaller surface area. Log yield was significantly lower for the Vimek and averaged 50% instead of 60% for the other treatments. Given that age, clone, stocking, and DBH were the same for all sample plots, it is unlikely that the different yields would point to different experimental conditions, a more logical explanation may lay with a different processing efficiency or with measurement errors on the part of the research team. The Vimek micro-forwarder hauled its payload over a significantly longer distance (380 m vs. 320 m; $p = 0.03$) than the Sampo thinning forwarder

because the nearest landing was occupied in November and the Vimek micro-forwarder had to unload its cargo 60 m further on.

On a sample plot basis, harvester productivity ranged from 2.2 to 4.2 BDT SMH$^{-1}$, and harvester cost from EUR 11 to 22 to BDT$^{-1}$. The Vimek harvester was significantly less productive than the rest (Table 3), which could be partly explained by its small size and power. On the other hand, the harvester cost was significantly lower for the Agama and the Vimek. The productivity margin enjoyed by the more powerful machines (i.e., Rottne and Sampo) was not large enough to offset their higher cost. Essentially, this part of the study showed that it does pay to choose a smaller and cheaper harvester because the tree size limitation prevails over machine power, and it does not allow the more powerful and productive machines to unleash their full potential.

**Table 3.** Machine productivity and cost (mean values).

| | | Agama | Rottne | Sampo | Vimek | *p* |
|---|---|---|---|---|---|---|
| Harvester productivity | BDT SMH$^{-1}$ | 3.4 [a] | 3.2 [a] | 3.0 [ab] | 2.6 [b] | 0.0040 |
| Harvester cost | EUR BDT$^{-1}$ | 13.6 [a] | 20.3 [b] | 20.0 [b] | 15.4 [a] | <0.0001 |
| Forwarder productivity | BDT SMH$^{-1}$ | 2.7 [a] | 4.2 [b] | 3.3 [c] | 2.5 [a] | <0.0001 |
| Forwarder cost | EUR BDT$^{-1}$ | 14.7 [ac] | 9.6 [b] | 12.5 [c] | 16.6 [a] | <0.0001 |
| Total cost | EUR BDT$^{-1}$ | 28.3 [a] | 29.9 [ab] | 32.5 [b] | 32.0 [ab] | 0.0041 |

Notes: BDT = Bone-Dry tons (0% water mass fraction); SMH = Scheduled machine hour, including a 20% delay component; different superscript letters on mean values on the same row indicate a statistically significant difference at the 5% level.

Forwarding productivity and cost ranged from 2.0 to 4.5 BDT SMH$^{-1}$ and from EUR 9 to 20 BDT$^{-1}$ on a sample plot basis. Statistical analysis pointed to a clear stratification of data, whereby forwarding cost was lowest for the Rottne and Sampo treatments and highest for the Vimek treatment.

In that regard, it is important to recall that the wood harvested with the Agama, Rottne, and Sampo harvesters was forwarded by the same operator with the same machine—the Sampo FR28 thinning forwarder. Therefore, the comparison between forwarders is reduced to the Vimek 610 4-t micro-forwarder versus the Sampo FR28 10-t thinning forwarder. That comparison might be partly biased by a 20% difference in mean extraction distance. For the rest, the general comparison between the Agama, Rottne, and Sampo treatments was not affected much by the forwarding machine, operator, or extraction distance, which were the same for those treatments. Furthermore, the Sampo thinning forwarder extracted its sample plots in a random sequence, which excluded any day or time of day effects on the results of the comparison between those three treatments.

In the end, total harvesting cost (from standing trees to logs and tops piled at the landing) ranged most commonly between EUR 26 and 36 BDT$^{-1}$ (interquartile range), with a grand mean at EUR 31 BDT$^{-1}$. Differences between treatments were relatively small, although statistical analysis showed a significant difference between the Agama (cheapest, at EUR 28 BDT$^{-1}$) and the Sampo treatments (most expensive, at EUR 32 BDT$^{-1}$).

Regression analysis confirmed the strong relationship between field stocking, harvesting cost (Equation (1)—Table 4), and log yield (Equation (5)—Table 4). In particular, as field stocking increased from 45 to 60 BDT ha$^{-1}$, total harvesting cost dropped by about 20% (Figure 3), and log yield grew by 20% (Figure 4).

**Table 4.** Results of the regression analysis.

| [Equation (1)] € BDT$^{-1}$ = a + b BDT ha$^{-1}$ + c Agama + dRottne | | | | | Adj. R$^2$ = 0.696 | | n = 33 | | | |
|---|---|---|---|---|---|---|---|---|---|---|
| a | *p*-Value | B | *p*-Value | c | *p*-Value | d | *p*-Value | | | |
| 56.190 | <0.0001 | −0.453 | <0.0001 | −5.574 | <0.0001 | −2.561 | 0.0036 | - | | - |

| [Equation (2)] BDT SMH$^{-1}$ Fell = a + b BDT ha$^{-1}$ + c Agama + d Vimek | | | | | | | Adj. R$^2$ = 0.944 | | n = 33 | |
|---|---|---|---|---|---|---|---|---|---|---|
| a | *p*-Value | B | *p*-Value | c | *p*-Value | d | *p*-Value | | | |
| 34.372 | <0.0001 | −0.272 | <0.0001 | −7.392 | <0.0001 | −4.529 | <0.0001 | - | | - |

| [Equation (3)] BDT SMH$^{-1}$ Extraction = a + b BDT ha$^{-1}$ + c Distance + d Agama + e Rottne | | | | | | | | Adj. R$^2$ = 0.753 | n = 25 | |
|---|---|---|---|---|---|---|---|---|---|---|
| a | *p*-Value | B | *p*-Value | c | *p*-Value | d | *p*-Value | e | *p*-Value | |
| 2.322 | 0.0050 | 0.016 | 0.2710 | 0.0001 | 0.6190 | −0.619 | <0.0001 | 0.7624 | <0.0001 | |

| [Equation (4)] BDT SMH$^{-1}$ Extraction = a + b Distance | | | | | Adj. R$^2$ = 0.020 | | n = 25 | | | |
|---|---|---|---|---|---|---|---|---|---|---|
| a | *p*-Value | B | *p*-Value | | | | | | | |
| 2.799 | 0.0038 | 0.020 | 0.4978 | - | - | - | - | - | | - |

| [Equation (5)] Log % = a + b BDT ha$^{-1}$ | | | | Adj. R$^2$ = 0.900 | | n = 25 | | | | |
|---|---|---|---|---|---|---|---|---|---|---|
| a | *p*-Value | B | *p*-Value | | | | | | | |
| 24.715 | <0.0001 | 0.686 | <0.0001 | - | - | - | - | - | | - |

Notes: BDT = Bone-Dry tons (0% water mass fraction); SMH = Scheduled machine hour, including a 20% delay component.

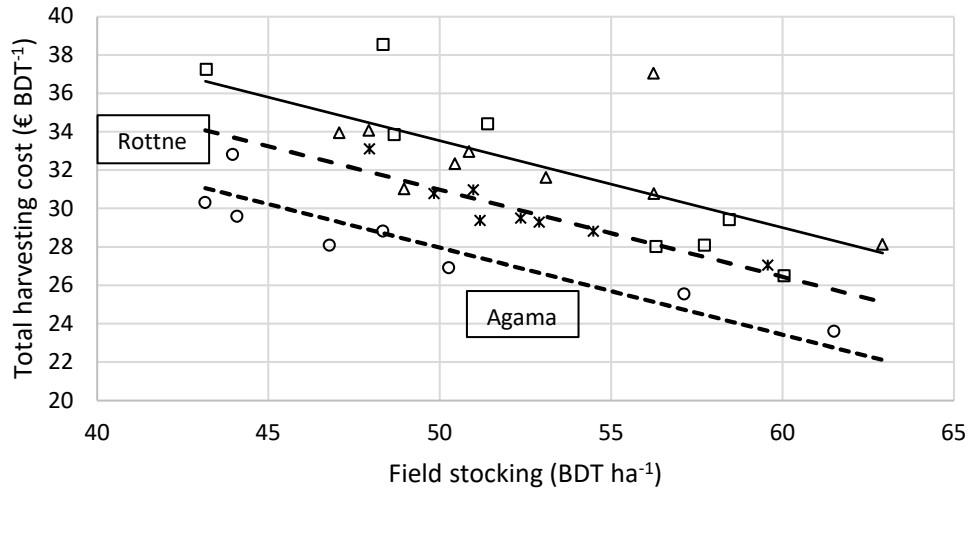

**Figure 3.** Total harvesting cost as a function of field stocking and harvesting chain: point scatter and regression graphs.

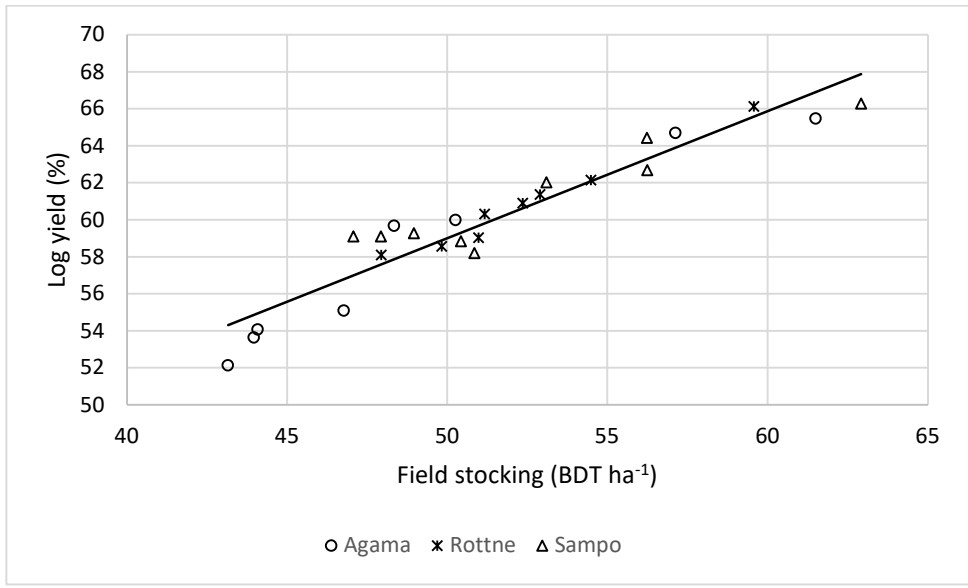

**Figure 4.** Log yield as a function of field stocking: point scatter and regression graphs (excluding Vimek chain).

Concerning log yield, it is important to stress that equation 5 was estimated after removing the Vimek harvester data since these were suspiciously different from the rest and may have been biased by some peculiar machine limitations or, more likely, by operator or researcher error. Once the odd data points had been removed from the dataset, the point cloud seemed consistent and aligned, and regression analysis offered a strong predictor.

A strong model was also obtained for harvester productivity (Equation (2)—Table 4), although taking one of the process steps in isolation from the others may downplay chain interdependency and provide a partial picture only. Nevertheless, this regression is useful for determining the effect of stocking on harvester performance and pinpointing the threshold below which single-tree handling might become a critical limitation. In turn, such knowledge may inform managers when deciding whether to stick with single-tree processing or move to multi-tree handling.

Finally, Equations (3) and (4) show that the effect of distance on extraction productivity is not significant: extraction productivity is only affected by treatment (Equation (3)). Obviously, that is true for the limited range of extraction distances gauged by this study. As a matter of fact, the range of distances was 250–400 m for the Sampo forwarder and 350–400 m for the Vimek forwarder. Apparently, that was not enough to have an effect on the productivity of the forwarders or to affect the comparison between the two machines. Eventually, the regression was conducted on the Sampo thinning forwarder data only (that is, Agama, Rottne, and Sampo treatments) because if we had introduced the Vimek micro-forwarder data, then the distance bias recorded for this treatment would have made it impossible to discriminate between treatment effect and distance effect (singularity error). Moreover, the range of distances spanned by the Sampo thinning forwarder was much wider than that covered by the Vimek micro-forwarder.

## 4. Discussion

As a start, it is best to address the main limitations of this study so that its findings are interpreted with due caution. The most severe limitation is that the test only included one stand type, one clone, and one single operator per machine. For that reason, readers must be warned about the risk of generalizing the results of this study, especially when facing operational conditions that present large deviations from those presented here. On the other hand, the current characteristics of most European medium-rotation poplar plantations are relatively homogeneous, and they are not widely different from those reported in this study. Probably, the most critical aspect was operator selection. While the operators selected for the study were considered sufficiently competent and widely representative by their employers, such evaluation was inherently subjective and did not guarantee that they were all equally competent. Therefore, if a slightly less competent operator ended up using a slightly better machine, that would obscure the eventual treatment difference and explain why this study did not find any major performance stratification between different machine treatments. Even in that case, the study would still offer a valid reference for the productivity of small-size CTL equipment deployed in European medium-rotation plantations, with a stocking between 40 and 65 BDT ha$^{-1}$.

The second caveat concerns the low log yield recorded for the Vimek treatment. In fact, working conditions were the same as for the other treatments, which offered comparable log yield figures among them. Furthermore, visual observation did not pick up anything peculiar in the work routine of the Vimek harvester.

In contrast, the forwarding distance handicap suffered by the Vimek treatment does not seem to represent a limitation of this study because statistical analysis showed that such a significant but relatively small difference was not large enough to affect forwarding productivity and cost and therefore could not invalidate the comparison presented in the study.

General corroboration is obtained when matching these results with those reported in previous studies. The very first choice falls on the most recent studies conducted by the same research team with the same methods under very similar work conditions (Table 5). After reconciling all figures to green tons, the results seem relatively well aligned, despite all differences deriving from variable sites, technology, and personnel conditions. Some of the differences may be explained by different stocking levels and/or harvesting methods. In particular, it is logical to expect a lower log yield from whole-tree harvesting operations, both because multi-tree processing may lead to a less rigorous value recovery and because whole-tree harvesting normally allows capturing larger amounts of biomass, which will dilute the percent representation of the log fraction. Abundant bibliography is also available on the harvesting of short-rotation poplar with single pass cut-and-chip technology, but that refers to the exclusive production of whole-tree chips from very young poplar and willow stands and does not compare well with the crops and harvesting chains described here [36].

**Table 5.** Comparison with the results of other recent studies on the same stand and operation types.

| Technology | Stocking | DBH | Felling | Processing | Extraction | Total Cost | Log Yield | Reference |
|---|---|---|---|---|---|---|---|---|
| type | gt ha$^{-1}$ | cm | gt SMH$^{-1}$ | gt SMH$^{-1}$ | gt SMH$^{-1}$ | EUR gt$^{-1}$ | % | |
| CTL | 120 | 12 | 6–7 | - | 5–10 | 12–14 | 50–60 | **This study** |
| CTL | 88 | 12 | 17–18 | - | 24 | 14 | 34–36 | [26] |
| WTH | 130 | 15–16 | 14 | 19 | 21 | 11 | 40–50 | [37] |
| WTH | 85 | 11.7 | 12–15 | 12–15 | 12–15 | 16–17 | 51 | [38] |
| WTH | 111 | 12.5 | 13–14 | 18–20 | 9 | 14–16 | 41 | [39] |

Note: all figures reconciled to green tons (gt); WTH = Whole-Tree Harvesting; CTL = Cut-to-Length.

That said, it is relatively safe to conclude from this study that larger thinning harvesters do not seem to enjoy any significant economic advantage over smaller micro-harvesters because their marginal productivity gain is obliterated by their higher operational cost [40]. If at all, some residual advantage may endure with their faster production rate, resulting in a shorter residence time that may help exploit a relatively short harvesting window—but that may be a relatively small benefit. On the other hand, micro-harvesters (and especially forwarders) would offer the advantage of lower ground pressure, which may represent a fundamental asset on those many wet sites where poplar is often planted. On the other hand, the smaller micro-harvester has lower operational flexibility than the (slightly) larger thinning harvester because it can only cope with very small trees—hence it can be effectively deployed on a smaller range of operations (short rotation poplar and first thinning, only).

If opting for a micro-machine, then one should consider the hybrid option because the power reserve accumulated in the batteries may be crucial to meet momentary surges in power demand that often turn into a "sticking point" when engine power represents a limitation. The study shows a dramatic difference in the performance of the two micro-harvesters; the hybrid Agama clearly outclassed the standard Vimek with a 30% increase in productivity. In fact, the lower productivity of the Vimek machine is explained in part by the need for this machine to resize the top material since the whole tops are too long (ca. 12 m total tree height—4 m log = 8 m top) for the Vimek micro-forwarder to haul efficiently. Therefore, the main limitation of the Vimek chain was in its forwarder component rather than the harvester—and the data in Table 3 clearly show that.

If a very poor soil bearing capacity does not impose resorting to a micro-forwarder, it may be more productive to deploy a larger thinning forwarder whenever possible. However, readers should be cautioned against extrapolating a theoretical lowest cost scenario by summing the lowest harvesting cost (Agama) with the lowest forwarding cost (Rottne) because such an exercise may generate an unrealistic prediction and lead to future disappointments. By definition, a chain is made of interconnected links, and the performance of any machine working in a chain is affected by the performance of the other machines working up and downstream from it.

The results of this study seem to confirm that the productivity of the thinning forwarder detached to support the Agama, Rottne, and Sampo harvesters was lowest when the forwarder was paired with the most productive harvester and higher when the forwarder was collecting from the other two less productive harvesters (Figure 5). Since the forwarder and its driver were the same, the sequence of extraction was randomized, and distance was too homogenous to have a significant effect, it is reasonable to attribute these differences in forwarder performance to different work site organization—likely stack size [41], alignment [42], layout, and orientation [43].

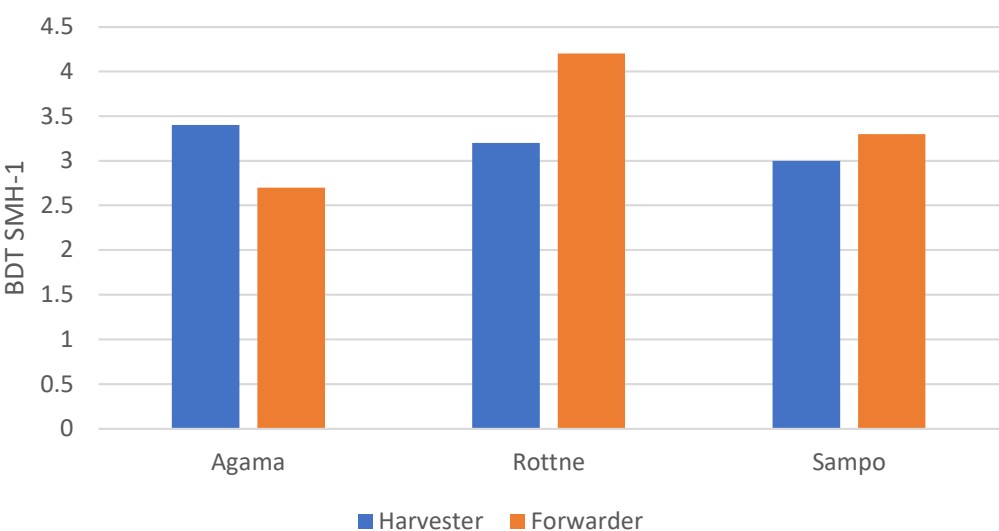

**Figure 5.** Harvester and forwarder productivity juxtaposed for the three chains using the same thinning forwarder (Sampo FR28).

Therefore, one coherent scenario involves the notion that the higher productivity of the Agama harvester was obtained by sacrificing some of the attention to stack organization, which would reflect a significantly lower forwarding performance. Of course, this is just a hypothesis that cannot be corroborated by any quantitative indicators since the study protocol did not include any measurements of stack quality. However, this conclusion is highly suggestive, and it draws attention to the fundamental issue of chain interdependency, which, although weaker than for other more complex harvesting systems, also affects CTL chains [44]. Similarly, this result highlights the need for measuring performance in terms of both work productivity and work quality, and not just productivity, since the same outfit under the same working conditions may achieve different productivity levels depending on how much attention is devoted to work quality [45].

The simple study protocol adopted for this study was easy to replicate [46] and was indeed able to hint at potentially intrinsic chain differences that did not depend exclusively on operator competence or whim or on work quality trade-offs, and it clearly described the major effect of field stocking on harvesting cost and log yield. This was expected since endless studies point to the close relationship between harvesting performance and tree size and in a plantation established at a set spacing, stocking closely reflects tree size [47–52]. The special merit of this study is that of producing a good estimate of the rate of increase that seems to be relatively suitable for generalization, given that data from all treatments on tests aligned to about the same slope (with non-significant variations). The study indicates that at a stocking of 60 BDT ha$^{-1}$, harvesting cost with one of the CTL chains on test will range from EUR 22 to 28 BDT$^{-1}$ (or EUR 10–12 per green ton), which may be well within the competitive price range for this crop type. However, less productive sites may be unable to offer such a stocking level, positioning somewhere between 40 and 50 BDT ha$^{-1}$. In that case, CTL harvesting cost will jump to EUR 30 to 36 BDT$^{-1}$ (or EUR 13–16 per green ton), which may prove critical. In such instances, managers may consider extending stand rotation and letting the trees grow bigger, if site fertility and legal constraints permit, or to shift to whole-tree harvesting, which is made more resilient to the effect of small tree size by its capacity for mass-handling. However, the productivity benefit of extending stand rotation should be balanced against its potential impact on regeneration since the ability to coppice decreases with stand age. Determining the potential of these options might be the goal of future research, so that plantation managers can be provided with a comprehensive toolbox.

A final consideration may be made about the potential of whole-tree harvesting to boost product recovery. Picking up the residue after stump-site processing leads to

inevitable losses, which likely explains why the correction coefficients found in this study were 0.99 for logs and only 0.72 for biomass. While error is inevitable, these results suggest that considerable losses are suffered with the biomass component, which corresponds to the visual observations of a substantial residue load after harvesting had been completed. While this component of the total harvest is still minor and carries a relatively small monetary value, it remains sizeable, and, what is more, it represents a hindrance to post-harvest site management activities, such as mechanical weeding and stump thinning. Therefore, properly managed whole-tree harvesting might represent an overall preferable solution for low-yielding plantations since it may dampen the negative effect of small tree size through mass handling, increase the amount of biomass being recovered, and facilitate post-harvest management.

## 5. Conclusions

The four alternative chains tested on 5-year-old poplar plantations have provided interesting results that may be generalized, although with some caution.

Total harvesting costs ranged between EUR 26 and 36 BDT. Statistical analysis showed a strong relationship between filed stocking, harvesting cost, and log yield.

The higher productivity gained with more powerful machines does not have any real economic benefit due to their higher costs. Furthermore, smaller machines have a lower ground pressure which is preferable in wet areas.

Finally, both work quality and work productivity should be considered when selecting one chain over the others.

**Author Contributions:** Conceptualization, R.S., N.M., B.K. and P.H.; methodology, All; formal analysis, R.S., N.M. and D.H.; resources, R.S., B.K. and P.H.; writing—original draft preparation, R.S. and N.M.; writing—review and editing, All; visualization, All; supervision, B.K. and P.H.; funding acquisition B.K., P.H. and R.S. All authors have read and agreed to the published version of the manuscript.

**Funding:** This study was funded by the Bio Based Industries Joint Undertaking under the European Union's Horizon 2020 research and innovation program under grant agreement No 745874 »Dendromass for Europe« (D4EU).

**Data Availability Statement:** Data can be obtained from the corresponding Author upon motivated request.

**Conflicts of Interest:** The authors declare no conflict of interest.

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
