# Peer review of "Cut-to-Length Harvesting Options for the Integrated Harvesting of the European Industrial Poplar Plantations"

_forests, doi:10.3390/f13091478_

Round 1

Reviewer 1 Report

Dear Editors, Dear Authors,

it is an honour to be considered as a reviewer for your manuscript “Cut-to-length harvesting options for the integrated harvesting of the European industrial poplar plantations “, submitted to forests. The manuscript (MS) is well written and provides insights into the productivity of different forest machines (and chains) during harvestings of poplar stands. Yet, after evaluating this work, I sensed quite some room for improvements, to match the standards of your journal. In the following, the main shortcomings and related suggestions are stated:

·        The MS is lengthy, and a significant amount of text is redundant

·        The MS’s structure is insufficient

·        Delimitation between main components, for example results and discussion, is not satisfying

·        The presentation of the results is imperfect.

In more detail:

Line

Comment

(50 -)57-72

Redundant, and kind off misleading for the reader. Frankly speaking, I would recommend to skip these lines, what applies to the following annotations of redundancy as well.

e.g. 113

“CTL” is not explained in captions. I am in favour of „stand-alone-captions “. In this regard, all captions could be more precise.

113, 133

Labels “A, B, C, D” could be used in Figures & captions

102

I strongly recommend to use sub-headings in the M&Ms

125

It seems pretty vague that “variability” is assessed using n=2. Maybe the formulation could be more cautious.

e.g. 127

Minor capitols are often written in major ones, e.g. “Cm” instead of “cm”. Guess this is the word-autocorrection. Throughout the manuscript, this occurred quite frequently. (“Ha”, “C” in the regression equations, …)

151 ff

The authors may consider to shift these findings into the results section. Out of my point of view, DBH and height were captured for the study and are results therefore. But this is a suggestion only.

184 ff

I think it is redundant to describe the multiple timing procedures. Rather the type of time study should absolutely be named here, also giving starting and end points of work elements.

220-222

Redundant? I think the discussion in 210-220 is no added value for the reader.

224 ff

Missing information. It is described that GLM was used. But a GLM without specifications is nothing but a LM. Also, post-hoc tests were done but this was not described here. And outliers were removed (i.e. Vimek, Figure 3), but this is described in results only.

241-247

This is clearly discussing the results à Discussion

250-256

Discussion

265-276

This paragraph could be rephrased. It is hard to follow the authors ideas.

286

Figure 3 is in a different style than Figure 5

288 f

Ok, here problems with Vimek data are described. And the reader knows, that footage was taken, but it is mentioned much later that it was considered therefore. A suggestion would be to skip the Vimek-data, since quite some problems are attributed to it. But I really want to leave that one to the authors of the MS

294

Equation 1 in bold?

295-302

Discussion

305-321

Discussion

340-355

No added value, I would say

361-382 (especially 370-376

Redundant. Actually, I think this could be said during a chat at some coffee break, but it does not belong into a scientific article. The MS should be slimmed down to results and discussion of the results.

424

Captions differ from figure. Costs vs. Productivity

437-446

Redundant

Shifting the discussing parts which has been inserted in the results to the actual Discussion would improve the MS vastly. Also, it would make a lot of the current Discussion obsolete.

Kind regards, a reviewer

Author Response

Dear Editors, Dear Authors,

it is an honour to be considered as a reviewer for your manuscript “Cut-to-length harvesting options for the integrated harvesting of the European industrial poplar plantations “, submitted to forests. The manuscript (MS) is well written and provides insights into the productivity of different forest machines (and chains) during harvestings of poplar stands. Yet, after evaluating this work, I sensed quite some room for improvements, to match the standards of your journal. In the following, the main shortcomings and related suggestions are stated:

  • The MS is lengthy, and a significant amount of text is redundant
  • The MS’s structure is insufficient
  • Delimitation between main components, for example results and discussion, is not satisfying
  • The presentation of the results is imperfect.

Dear Reviewer!

Thank you very much for your kind words, and for the many useful suggestions for improvement. My co-Authors and I appreciate your time and effort, and we are grateful for your valuable contribution to improving our manuscript. Please see below for our detailed answers to each individual comment.

Best regards

Natascia Magagnotti

Point 1: Line (50 -)57-72         Redundant, and kind off misleading for the reader. Frankly speaking, I would recommend to skip these lines, what applies to the following annotations of redundancy as well.

Response 1: We are not sure the paragraph is redundant or misleading: it would be redundant if repeated information already provided, and misleading if it led the reader in a different direction than intended, whereas our narrative is not repetitive and it does lead the reader towards the intended logical developments. However, we agree that it does not contribute unavoidable background information, and therefore we have reduced it significantly, based on the known principle that “less is more” ?

Point 2: Line e.g. 113  “CTL” is not explained in captions. I am in favour of „stand-alone-captions “. In this regard, all captions could be more precise.

Response 2: Edited as suggested

Point 3: Line 113, 133 Labels “A, B, C, D” could be used in Figures & captions

Response 3: That would be a good solution, but the system we have used seems equally effective as the one suggested by the Reviewer – or at least not dramatically less effective. We feel that changing all headings and captions would cost more additional work than the actual benefit it would generate, going against the efficiency principle…this study is about efficiency, after all, and we should be intellectually consistent ?

Point 4 Line 102 -        I strongly recommend to use sub-headings in the M&Ms

Response 4: That is a very good idea. We have introduced the sub-headings as suggested

Point 5: Line 125 -       It seems pretty vague that “variability” is assessed using n=2. Maybe the formulation could be more cautious.

Response 5: We have reformulated the sentence, as suggested

Point 6: Line e.g. 127 - Minor capitols are often written in major ones, e.g. “Cm” instead of “cm”. Guess this is the word-autocorrection. Throughout the manuscript, this occurred quite frequently. (“Ha”, “C” in the regression equations, …)

Response 6: That naughty autocorrect function! We have checked the tables and graphs and restored consistency…

Point 7: Line 151 ff      - The authors may consider to shift these findings into the results section. Out of my point of view, DBH and height were captured for the study and are results therefore. But this is a suggestion only.

Response 7: I agree with the Reviewer: both options are viable…but I’ll stick to the principle of maximum efficiency of not doing a work I am not sure is going to significantly improve the product! ?

Point 8: 184 ff - I think it is redundant to describe the multiple timing procedures. Rather the type of time study should absolutely be named here, also giving starting and end points of work elements.

Response 8: I am not sure…We set up a redundant system for a reason (security against data loss) and that system is useful enough that it deserves description for possible emulation by colleagues, especially if the data they collect are very valuable and they cannot afford a repeat of the test…Besides, the article is electronic, so we are not wasting paper anyway! ? Concerning the time elements, their description would apply to a cycle-level elemental time study. In fact, this study is plot-level (not cycle-level) and is not elemental, so there are no work elements in it. Of course, we did record separately productive work time and delay time, but a clear description of both is contained in the seminal publication we have duly quoted.

Point 9: Line 220-222 - Redundant? I think the discussion in 210-220 is no added value for the reader.

Response 9: The statement is not void and it may have its place, but I agree with the Reviewer that “less is more”. So, we have removed it.

Point 10: Line 224 ff - Missing information. It is described that GLM was used. But a GLM without specifications is nothing but a LM. Also, post-hoc tests were done but this was not described here. And outliers were removed (i.e. Vimek, Figure 3), but this is described in results only.

Response 10: I am not sure I understand some of the three issues described in this comment, so I’ll go one by one: a) I believe you need specifications to describe a Generalized Linear Model, not a General Linear Model; b) we have introduced a description of the post-hoc test as suggested; c) yes, we could define that as removing outliers, but in fact we removed a whole treatment from that analysis, so I am not sure that defining such procedure as “outlier removal” would correctly describe what we did. At any rate, we did describe what we did at the pertinent point in the narrative, so readers are informed of what we have done. 

Point 11: Line 241-247 -          This is clearly discussing the results à Discussione

Response 11: We agree with the Reviewer that the text in the prapagraph is indeed more than just a pure description of the equations presented in the Table. On the other hand, we believe that an inflexible separation between pure description of the results and discussion of their meaning may be inefficient. Immediate implications of the results should be discussed as soon as the results are presented: readers will do that in their minds anyway – it is human nature – and we if postpone those immediate inferences to later on, then we risk losing the reader’s attention…to the only benefit of diluting the later discussion. In contrast, the discussion section should focus on the deeper meaning and the strategic implications of the study results. From a communication viepoint, we believe that is way more effective than the strict separation advocated by some of our colleagues. Maybe that is the tradition, but traditions must evolve in order to increase efficiency – and as scientists we must be the first ones to believe in progress and support evolution…? 

Point 12: Line 250-256             - Discussion

Response 12: Please see above

Point 13: Line 265-276             This paragraph could be rephrased. It is hard to follow the authors ideas.

Response 13: We have rephrased the partagraph as suggested, and we hope its meaning is now clearer.

Point 14: Line 286       Figure 3 is in a different style than Figure 5

Response 14: I am sorry: I must have interpreted the comment incorrectly, but I do not know how else I must read it. I have checked the type format and size, but they are indeed the same for both figures. They are both Excel-drawn graphs. Obviously they are different types of graphs, but the style seems to be the same. In case, please direct me to the inconsistencies and I’ll be happy to solve them.

Point 15: Line 288 f- Ok, here problems with Vimek data are described. And the reader knows, that footage was taken, but it is mentioned much later that it was considered therefore. A suggestion would be to skip the Vimek-data, since quite some problems are attributed to it. But I really want to leave that one to the authors of the MS

Response 15: We are not sure about how we should interpret that comment. What we did was to integrate the Vimek time and production data but exclude the log yield data. So, if the Reviewer suggests removing the log yield data, we have done it already; on the other hand, if he/she suggest removing the Vimek all together from the study, we would object. We did collect the time data, although from a video and not in person. But the researchers performing the time study were the same in all cases and – as we said in the text – the time study protocol was specifically designed to be simple and capable of being easily replicated by any researchers with limited risks for systematic deviations. So, the time and productivity data are most likely correct.  

Point 16: Line 294       Equation 1 in bold?

Response 16: Some glitch with autoformat: sorry for that…MS Word tends to be a bit more independent than we would like it to be…We have correct the format accordingly

Point 17: Line 295-302             Discussion

Response 17: Please see Response 11

Point 18: Line 305-321             Discussion

Response 18: As above

Point 19: Line 340-355             No added value, I would say

Response 19: We have removed the paragraph. As they say: less is more…

Point 20: Line 361-382 (especially 370-376) Redundant. Actually, I think this could be said during a chat at some coffee break, but it does not belong into a scientific article. The MS should be slimmed down to results and discussion of the results.

Response 20: We are not sure that all the paragraph is redundant, so we have shortened it  (removing especially lines 370-376) but not entirely removed it. We would also like to clarify two things: 1) we wanted to justify why we cited only our own work, given that some people may find that suspicious or objectionabl; we did try hard to find something comparable and we did not find it (I assume that if we had ignored something relevant the Reviewer would have objected to self-citing for the sake of it, rather than to the text per se)  and 2) we believe that scientific papers should start to use colloquial language more often - provided the contents have scientific value, of course…the time of white-bearded boring scientists uttering incomprehensible statements must end ?

Point 21: Line 424 - Captions differ from figure. Costs vs. Productivity

Response 21: Of course! An oversight, obviously…Thank you for pointing us at it. We have edited the caption as required.

Point 22: Line 437-446 - Redundant

Response 22: We have removed the paragraph, as suggested. A more concise paper is always better than a lengthy one!

Reviewer 2 Report

Authors presented results of a high practical value in the field of forest engineering supported by a very interesting discussion.

In my opinion only few corrections and clarifications (listed below) are needed prior to publication:

L78-79: Statement should be supported with reference.

L106: Were only tops forwarded or also branches originating from trunk processing?

L110: I suggest “cutting diameter” instead of “cutting capacity”.

L199-204: Please clarify the forwarding time consumption of partial loads regarding the loading and unloading time consumption.

L281-284: Was the filed stocking difference caused by different DBH or number of trees per ha? If DBH distribution caused the stocking difference it could be assumed that the log yield (due to top diameter threshold) is strongly influenced by the DBH rather than the stocking. Was only one 4 m log produced per tree in all the cases as suggested by the L404?

L286: Figure 3. Presents total harvesting cost vs. field stocking not the Log yield; Figure 4. Is not presented in the manuscript – please check

Table 4: please check the Eq. 2 Caption – does the equation present the productivity (BDT/SMH) or cost(€/BDT)?

L303-321: Please describe in more detail equations 3 and 4 separately. Table 4 captions of Eq. 3 and Eq.4 indicate that the productivity difference was tested while in line 315 cost is mentioned. Please state the extraction distance span of Agma, Rottne and Sampo treatments and Vimek treatments. Can the result of extraction distance vs. forwarding productivity in the case of Sampo forwarder (if limited variation of extraction distance exists) substantiate the assumption that a 60 m extraction difference between the tested forwarders had no effect on the productivity? If the extraction distances of the Sampo forwarder are of limited variation and the extraction distance range (min-max) of the two forwarders in question does not partially overlap, I suggest commenting the forwarding results of the Vimek and other tested systems without direct comparison.

Author Response

Authors presented results of a high practical value in the field of forest engineering supported by a very interesting discussion.

In my opinion only few corrections and clarifications (listed below) are needed prior to publication:

Dear Reviewer!

Thank you very much for your favorable appreciation of our work and for your valuable insights offered through your comments. My co-Authors and I appreciate your work, and we are grateful for your help with improving our manuscript. Please see below for our detailed answers to each individual comment.

Best regards

Natascia Magagnotti

Point 1:L78-79: Statement should be supported with reference.

Response 1: We have introduced a supporting reference, as suggested

Point 2: L106: Were only tops forwarded or also branches originating from trunk processing?

Response 2: Good point. We took it for granted that “tops” would also include the major recoverable branches, but actually it does not…we have rephrased the sentence with a better description.

Point 3: L110: I suggest “cutting diameter” instead of “cutting capacity”.

Response 3: Indeed. Edited as suggested.

Point 4: L199-204: Please clarify the forwarding time consumption of partial loads regarding the loading and unloading time consumption.

Response 4: The loading and unloading time were the actual ones and they were not modified in any way. If loading time is proportional to the mass being loaded, then it does not matter whether the machine is loading a full payload or not: the loading time per m3 (or ton) will be correct anyway. The same accounts for unloading. The issues arises with travel time, since that is only slightly affected by load size – in fact, if the load is not excessive, travel time will only depend on distance; for the same distance, travel time per ton will increase with decreasing payload. Therefore, if we artificially decrease payload for study reasons we will get an unrealistic estimate of travel time per m3 (or ton). To avoid that, we must also correct travel time per m3 (or ton) proportionately. That is what we have done and we feel that is well described in the manuscript as follows ”Therefore, the vehicle travel time was multiplied by the payload fill rate of that load in order to account for the fact that a full load would have been extracted in real operational conditions, and therefore that same travel time would have been distributed over a larger payload”. We have not mentioned the treatment of loading or unloading time because they were not altered in any way, so it felt useless to describe something we did not do.

Point 5: L281-284: Was the filed stocking difference caused by different DBH or number of trees per ha? If DBH distribution caused the stocking difference it could be assumed that the log yield (due to top diameter threshold) is strongly influenced by the DBH rather than the stocking. Was only one 4 m log produced per tree in all the cases as suggested by the L404?

Response 5: Thank you for the suggestion: that is a very good point, which we are going to further explore using the data from this study and those from other current and future studies we are conducting (or plan to conduct) in the same plantations. The field stocking difference is generally a combination of both the number of trees and their size (DBH), since both growth rate and mortality are affected by local differences in soil fertility, water availability etc. Therefore, it may be difficult to isolate the effect or one from that of the other. Besides, the operators were instructed to go to for an 8 cm top diameter and if the tree was large enough they would indeed produce two logs, although that was not very frequent in this field.

Point 6: L286: Figure 3. Presents total harvesting cost vs. field stocking not the Log yield; Figure 4. Is not presented in the manuscript – please check

Response 6: Thank you very very much for pointing that out! It was a huge mistake, probably caused by some editing we did before submitting the paper. The graph was that for Figure 3 but the caption was that for Figure 4, which was missing altogheter. Huge mess-up, and one of those you cannot see when it is in front of your face! Thanks for detecting it. We have reintroduced Figure 4 and fitted the right captions to each of the two figures…

Point 7: Table 4: please check the Eq. 2 Caption – does the equation present the productivity (BDT/SMH) or cost(€/BDT)?

Response 7: Of course! Another serious mistake, and this time it was not MS Word’s autocorrect function! We checked the original file and there was a transcription error, when writing Minitab’s output into Excel. Sorry for that…and thank you very much again for spotting something quite important!

Point 8: L303-321: Please describe in more detail equations 3 and 4 separately. Table 4 captions of Eq. 3 and Eq.4 indicate that the productivity difference was tested while in line 315 cost is mentioned. Please state the extraction distance span of Agma, Rottne and Sampo treatments and Vimek treatments. Can the result of extraction distance vs. forwarding productivity in the case of Sampo forwarder (if limited variation of extraction distance exists) substantiate the assumption that a 60 m extraction difference between the tested forwarders had no effect on the productivity? If the extraction distances of the Sampo forwarder are of limited variation and the extraction distance range (min-max) of the two forwarders in question does not partially overlap, I suggest commenting the forwarding results of the Vimek and other tested systems without direct comparison.

Response 8: We have described the equations in more detail and clarified the statement at line 315. We have also provided information about the extraction distance span for all treatments. They do overlap, but the Vimek worked on longer distances as an average (range 350-400 m vs. 250-400 m for the others). We have also rephrased the statement about the effect of extraction distance differences.

Round 2

Reviewer 1 Report

Dear Editor, Dear Authors,

out of my point of view, the manuscript experienced a substantial improvement. Although I do not share the opinion stated in Response 11, I see the point and it is to the authors discretion how to handle the separation of results and discussion.

One additional note: in Figure 4, the labels shown in the legend are incomplete. Also, Figure 3 has a solid frame line, which is missing Figures 4 & 5. For the overall appearance, I would suggest that the Figures are edited.

Minor editing (e.g. hyperscript of affiliations, L47, L50, ...) will be handled by the proof-reading of the journal I suppose.

Kind regards, a reviewer

Author Response

Point 1 - out of my point of view, the manuscript experienced a substantial improvement. Although I do not share the opinion stated in Response 11, I see the point and it is to the authors discretion how to handle the separation of results and discussion.

 Response 1 – Thank you very much. We appreciate everyone has specific preferences: there is no “one size fits all” approach. We also acknowledge that our preference is not any better than that of the Reviewer, or of any other similarly competent expert. It is just our own preference: as Authors we are entitled to it, but we foster no illusions that it is superior to the others…

Point 2 - One additional note: in Figure 4, the labels shown in the legend are incomplete. Also, Figure 3 has a solid frame line, which is missing Figures 4 & 5. For the overall appearance, I would suggest that the Figures are edited.

Response 2 – Thank you for directing us at the specific issues afflicting the figures. We have now spotted them, and we have edited all inconsistencies accordingly

Point 3 - Minor editing (e.g. hyperscript of affiliations, L47, L50, ...) will be handled by the proof-reading of the journal I suppose.

Response 3 – Of course. One can never get all formatting details right, anyway…and if one does, Word autocorrect will make it sure to infiltrate something odd…?